# Bromatological Analysis and Characterization of Phenolics in Snow Mountain Garlic

**DOI:** 10.3390/molecules27123712

**Published:** 2022-06-09

**Authors:** Yolanda Terán-Figueroa, Denisse de Loera, Alberto Toxqui-Terán, Gabriela Montero-Morán, María Zenaida Saavedra-Leos

**Affiliations:** 1Faculty of Nursing and Nutrition, Autonomous University of San Luis Potosí, Niño Artillero Avenue #130, University Zone, San Luis Potosí C.P. 78240, Mexico; yolandat@e-uaslp.mx; 2Faculty of Chemical Sciences, Autonomous University of San Luis Potosí, Dr. Manuel Nava Martínez Avenue #6, University Zone, San Luis Potosí C.P. 78210, Mexico; atenea.deloera@uaslp.mx (D.d.L.); gabriela.motero@uaslp.mx (G.M.-M.); 3Advanced Materials Research Center (CIMAV), Alianza Norte 202, Research and Technological Innovation Park (PIIT), Apodaca C.P. 66600, Mexico; alberto.toxqui@cimav.edu.mx; 4Coordinación Académica Región Altiplano, Universidad Autónoma de San Luis Potosí, Carretera Cedral Km, 5+600 Ejido San José de las Trojes, Matehuala C.P. 78700, Mexico

**Keywords:** analysis, garlic, composition, antioxidant activity, ashes

## Abstract

The remarkable properties of garlic *A. sativum* L. have been described, but little is known about Snow mountain garlic. Understanding general aspects of this garlic composition, including the presence of phenolics, will establish its possible use for health or infer which compounds can contribute to improving it. This study aimed to determine the ash content, lipid profile, and characterization of phenolics in Snow mountain garlic. The organic content was obtained by common techniques (oven drying, calcination, Kjeldahl method, etc.). The quantitative analysis of the ashes was made by Inductively Coupled Plasma Emission Spectrometry. The fatty acid profile was determined by Gas Chromatography. The presence of phenolics was determined by foam, Libermann–Burchard, Dragendorff, Salkowski, ferric chloride, vanillin, catechin, Constantinescu, and Shinoda reactions. The total phenolic content was determined via the Folin–Ciocalteu method, and antioxidant activity was determined using the DPPH radical method. The bromatological analysis showed a 51.1% humidity, and the main organic compounds were carbohydrates (46.7%). Ash analysis showed 287.46 g/kg of potassium. The fatty acid profile showed 75.61% of polyunsaturated fatty acid. Phenolics like saponins, alkaloids, triterpenes, tannins, and flavonoids were present. Antioxidant activity was found by radical DPPH of 25.64 (±0.78) µmol TE/1 g dw. Snow mountain garlic shares a composition similar to those found in other garlic.

## 1. Introduction

The family Alliaceae comprises approximately 850 species characterized by underground bulbs [1]. Allium species have been used for centuries for their flavor as vegetables and ethnomedicine to prevent various diseases. Members of this genus are known for their sulfur compounds [2,3]. *Allium sativum* L. garlic (common garlic) has been used as a food and component of multiple medicinal preparations since ancient times. Scientific interest in this plant dates to the last century when it was observed that the compounds extracted from garlic could effectively inhibit the growth of various strains of fungi, highlighting allicin—a sulfur compound—and for its antimicrobial and antifungal effects [4]. According to numerous clinical trials, it effectively prevents and treats atherosclerosis and blood pressure as an antiplatelet agent; it also has antioxidant, lipid-lowering, antiatherogenic, anticarcinogenic, and immunomodulatory activity [5,6,7].

Plants of the genus *Allium* have a high content of minerals, essential amino acids, fiber, vitamins, flavonoids, and phenolic compounds. Among the important secondary metabolites of the plants of this genus are the organosulfur compounds, which are responsible for their bioactive properties. The bioactivity of alliaceous plants is mainly due to compounds derived from thiosulfinates [8].

Just as the excellent properties of *A. sativum* L. have been described, little is known about its cousin “Snow mountain”, also called “Kashmiri garlic” or “Himalayan garlic”. This is considered a purified form of garlic. In ancient times, this garlic was used by mountaineers to raise their energy levels and detoxify their bodies in extreme cold weather conditions. Local people generally consume it as a remedy for rheumatoid arthritis and other promising medicinal properties. The curative effects against hypertension, atherosclerosis, diabetes, cancer, and particular immunomodulatory activity has been reported [9,10]. The benefits of *A. sativum* L. result primarily from the composition and concentrations of fatty acids, antioxidants, and flavonoids, among others [11,12]. Knowing general aspects of the composition of this plant will allow to establish its possible use for health or, to infer which compounds are those that can contribute to improving it. Then, this study aimed to determine the ash content, lipid profile, and characterization of phenolics in Snow mountain garlic.

## 2. Results

Snow mountain garlic produces a single clove [9] surrounded by a hardcover with a flat and convex portion, a pointed end, and a blunt end. Inside the shell is a white, orbed, hydrated onion-shaped structure (Figure 1). In 100 g, there are approximately 146 garlic cloves of different sizes (Table 1).

The bromatological analysis showed a humidity of 61.5% and a water activity (a_w_) of 0.98. The pH was 6.5 (Table 2), the protein concentration was 698.1 µg/mL, and the concentration of reducing sugars could not be determined with the methods used. The organic compounds in greater quantity were carbohydrates (46.7%). The ashes correspond to fiber, vitamins, and minerals (1.2%) (Table 2).

The ash analysis showed a higher percentage of potassium, phosphorus, and magnesium (287.46, 91.58 and 21.20 g/kg, respectively). Traces of arsenic, cobalt, lead, and cadmium were found (Table 3).

Regarding the fatty acid profile, the content of 3.12% and long-chain fatty acids range from 10 to 20 C. It was also possible to identify both saturated (20.15%) and monounsaturated (4.24%). In a higher percentage, polyunsaturated were found (75.61%) (Table 4).

The phytochemical analysis was determined on hexanic, chloroformic, and ethanolic extracts. The results indicate that metabolites present in Snow mountain garlic are of medium to high polarity since all metabolite types tested were founded in chloroformic and ethanolic extracts, being higher in the ethanolic extract. On the other hand, the hexanic extract only showed the presence of saponins and tannins (Table 5).

The total phenolic content (TPC) showed 2.19 (±0.18) mg GAE/g dw and antioxidant activity of 25.64 (±0.78) µmol TE/g dw.

## 3. Discussion

The initial works that help to elucidate the possible bioactive components of a plant are based on the determination of the presence of macromolecules in percentages, as well as other general characteristics. These data also serve to establish the nutritional contribution. According to the data obtained in the development of the present work, it was possible to determine the percentage of the essential macromolecules of a bromatological analysis: carbohydrates, lipids, proteins, humidity, a_w_, and ashes. Water is the main component in the Snow mountain garlic. When comparing the results with those reported for common garlic, values vary, showing higher values of moisture percentage (61.5%) compared with common garlic, 58.58% from 100 g of fresh garlic [11]. However, humidity values could vary according to the growing region depending on the genotype [13]. Moisture to seven different varieties of fresh garlic (*Bjetin*, *Vekan*, *Havel*, *Ivan*, *Rusák*, *Havran*, *Lukan)* has been determined. The authors found a range of 65 to 57% humidity; however, they do not report the a_w_ of every garlic studied [14,15]. Moisture content and a_w_, in addition to temperature, are the parameters that most affect the rate of deterioration reactions in foods. Water activity describes the limits of bound water contained in a food and its availability to act as a solvent and participate in chemical reactions [16].

In second place, carbohydrates were positioned with 46.8%, however, 20–30% has been reported for common garlic [17]. Another investigation reports between 23 to 36% of total carbohydrate content [18], which has the highest percentage value of all the macromolecules determined in Snow mountain garlic and *A. sativum* L. garlic. Carbohydrates are essential for the structure of cells and for obtaining energy; hence they are abundant. These are commonly present as sugars; therefore, they represent the primary energy source for garlic and those who consume it.

Regarding lipids, it was found that Snow mountain garlic has a minimal amount (less than 1%), like *A. sativum* L. and black garlic, which contain 0.3% [19] and 0.5% [20], respectively.

These macromolecules are of great importance due to their various biological functions. For example, they are structural components of cell membranes. Some have important biological activity as they are precursors of fat-soluble vitamins. In plants, they cover the surface of the stems, leaves, and fruits since they prevent excessive water evaporation and protect against attacks by insects and parasites [20].

Regarding proteins, a value of 0.547% indicated that it is not a very important source of nutrients or components in the plant. However, when comparing it with that reported for common garlic, it was found that common and black garlic have higher amounts, 4–5 [21] and 17% [19], respectively

The ashes content in Snow mountain was similar to *Allium sativum* L. garlic, 1.2%, and 1.1–1.8%, respectively. On the other hand, it was twice less than reported for black garlic (3.5%) [13,19]. It is important to remember that black garlic is the product of subjecting common garlic to high temperatures.

Regarding the composition of the ashes, it is reported that *A. schoenoprasum* L. (known as chives) contains a priority mineral potassium with 296 mg/g, followed by calcium, phosphorus, and magnesium at 95, 58 and 42 mg/g, respectively. In small quantities, copper and zinc were found with 0.56 and 0.157 mg/g and 0.9 µg/g of selenium [22]. In another job it was found copper, zinc, selenium, arsenic, cobalt, nickel, lead, mercury, and cadmium elements. Potassium, as a macronutrient, plays essential roles in different and diverse physiological processes during development and plant growth [23]. The same function has been attributed to the two most abundant divalent cations in plants, phosphorus, and magnesium, in addition to the role of calcium in cell signaling. Antagonistic interactions in plant cells depend on a homeostatic balance between these cations [24]. Phosphorus has been described as the most important macronutrient and is especially related to crop productivity [25]. Generally, heavy metal micronutrients such as Co, Cu, Fe, Mn, Mo, Ni, and Zn play an important role in plant cell growth and development [26]. Zn, as in animals, actively participates as a cofactor of many enzymes that regulate transcription activities and signal transduction, among others [27].

Another result obtained was the pH value. Although it is not part of nutritional analysis, it was considered important because it can be useful data in the therapeutic application if a pharmaceutical formulation were made with this garlic. The result indicates an acid pH (6.5), like *A. sativum* L., which has a pH of 6.05 [28]. Our laboratory determined the pH of common garlic under the same conditions established for Snow mountain, obtaining a value of 6.3, very similar to the reported. It is known that excessive garlic consumption, as a condiment or in traditional medicine, can cause adverse effects in humans, including a burning sensation in the mouth and gastrointestinal tract and nausea, among others [29], where pH may be playing a significant role.

Rukmankesh et al. (2020) mentioned that *A. schoenoprasum* L. is also known as Snow mountain garlic or Kashmiri garlic [9]; however, in our experience, the flowers are different from this garlic. The fatty acid profile has been previously analyzed in *Allium sativum* L. garlic; the characterization was made in two varieties of common garlic (red and white), harvested in different regions of Italy. Using HRMAS-NMR (High-resolution magic angle spinning-nuclear magnetic resonance) spectroscopy, the most abundant components in this garlic were identified, in which the fatty acids present in greater quantity were linoleic acid (C18:2) and palmitic acid (C16) [30].

On the other hand, Kamanna et al. (1980) performed an extraction with chloroform and methanol of the lipids in common garlic for later fractionation by column chromatography of silicic acid. The fatty acid composition of total lipids showed that they were linoleic acid (C18:2) with 64.8% of total lipids, palmitic (C16:0) with 24%, linolenic (C18:3) with 5.7%, and oleic (C18:1) with 3.1% [31].

In Snow mountain garlic, the percentages of fatty acids were lower; for linoleic acid 10.93%, palmitic acid 4.22%, linolenic acid 1.69%, and oleic acid 3.10%. The fatty acids with higher rates were hexadecane and hexadecatriene with 53.5%. Hexadecadieneic acid is a rare fatty acid that has been identified in some species of marine sponges. Caballeira et al. (2002) report that this fatty acid has moderate antimicrobial activity against Gram-positive bacteria *Staphylococcus aureus* (MIC 80 µM) and *Streptococcus faecalis* (MIC 200 µM). This acid is ineffective in inhibiting Gram-negative bacteria such as *Pseudomona aeruginosa* and *Escherichia coli.* However, hexadecanoic acid (16:0) does not show activity against any of these microorganisms, which shows that the double bonds within hexadecadieneic acid are necessary for antimicrobial activity. This acid can inhibit topoisomerase I at a concentration of 800 µM. Still, with a lower potency than long-chain fatty acids (C-27-C-28), it is said that the length of the carbon chain plays an important role in the inhibition of this enzyme [32]. On the other hand, Yuko et al. (2010) identified in radish plants (*Raphanus sativus* L.) hexadecatrienoic acid as an anti-bolting agent (premature flowering) [33].

The major constituents of all plant cells, which are also essential, are the fatty acids and, in general, the lipids, which provide energy for metabolism and participate as signal transduction mediators. Lipids and fatty acids act as intracellular and extracellular signals [34]. Determination of fatty acid esters of hydroxy fatty acids (FAHFAs) in Snow mountain garlic is important due to their effect on human health, such as anti-diabetic and anti-inflammatory activities, reduction in adipose tissue and serum of insulin-resistant humans, reduction of inflammatory cytokine production, and in adipose inflammation in obesity [35,36].

One of the most representative stages in the preliminary study of a plant species is the phytochemical analysis that helps understand its physiology and biochemistry and achieve its best scientific and medicinal purposes. The results of Snow mountain garlic analysis indicated the presence of secondary metabolites of medium to high polarities such as flavonoids, alkaloids, saponins, and tannins.

Epidemiological studies suggested that consumption of diets rich in plant polyphenols in the long term offered some protection against diabetes, development of cancers, osteoporosis, cardiovascular and neurodegenerative diseases [37], enzyme inhibitory activities like hydrolases and cyclooxygenases, among others, and anti-inflammatory, anticancer, antibacterial and antiviral action [38]. The beneficial effects of vegetables and fruits are well known and diffused among the populations and are attributed to bioactive compounds, like flavonoids, phenolic acids, lignins, and stilbenes, polyphenols that exhibit antioxidant effects. Although the identification of organosulfur compounds in *A. sativum* L. and their functions concerning human health has been widely described [39], the presence of antioxidants has not been studied.

Locatelli and collaborators demonstrated that raw garlic showed the highest antioxidant activity by free radical scavenging against 2,2-diphenyl-1-picrylhydrazyl radical (DPPH), 2,2′-azino-bis (3-ethylbenzene-thiazoline-6-sulfonic acid) diammonium salt (ABTS+), and Fe(III) reducing ability (FRAP). The garlic antioxidant activity could be due to allicin, by an antiradical action mechanism, and the phenolic compounds known as antioxidants [40].

Total phenolics content (TPC) varied widely in the genus *Allium*. In garlic, *Allium sativum* L. was 75–700 mg GAE/kg, different from what was found in Snow mountain garlic with 2.19 (±0.18) mg GAE/1 g. On the other hand, it is reported that the TPC was increased by about 4–10-fold in the black garlic cloves compared with the fresh garlic [41,42,43]. In this work, we identified the presence of alkaloids with significant antioxidant activity in an ethanolic and methanolic aged garlic extract of Snow mountain garlic. Recently, the phenolic profiles of methanol extracts of *A. ampeloprasum* L. (elephant garlic) have been described. Among 30 phenolic compounds were quantified. Bioactive flavonols, quercetin (98.37 ± 1.75 µg/g extract), and kaempferol (173.20 ± 1.3 µg/g extract) have been reported in garlic [44]. Therefore, the composition and concentrations of metabolites vary depending on the garlic specie.

## 4. Materials and Methods

### 4.1. Garlic Material

The Snow mountain garlic was obtained from a local market (Reforma street 405A, Historical Center, 78000 San Luis Potosí, Mexico) and was identified by the local herbalist. We acquired a batch of 2 kg to develop the present work.

### 4.2. Obtaining the Size of Garlic Cloves

100 g of garlic cloves were weighed in triplicate. By approximate size (measured in cm of height, width, and base), they were separated into three groups: small, medium, and large. The measurements were done as follows: height from the flat portion to the concave at its longest part, the width from the midpoint between the flat portion to the concave portion on the transverse axis, and the base (flat part) by measuring from the pointed end to the blunt end by the flat part. The average of the three measurements was obtained.

### 4.3. Obtention of Organic Content

Water activity (a_w_) was determined in an Aqualab Series 3 apparatus (Decagon Devices, Inc., Pullman, WA, USA). The water content (%) was obtained according to the AOAC method, drying the sample in an oven at 110 °C for 2 h [45]. The percentage of ashes was obtained by the calcination method [45]. The Kjeldahl method determined the crude protein content [46] in a Kjeldahl micro-distiller (SEV-Prendo, model DEK, Mexico). Crude fat was obtained following the Soxhlet method [47]. The percentage of total carbohydrates was obtained by the acid hydrolysis method [48]. The Nelson–Somogyi method was used to obtain the concentration of reducing sugars [49]. The pH was determined in the same extract with a potentiometer (Ohaus model Starter 3100). Each determination was made in duplicate.

### 4.4. Extraction and Analysis of Ash

#### 4.4.1. Obtaining Ashes

The percentage of ashes was obtained by the calcination method [45]. Briefly, a crucible was placed at 600 °C for 2 h until a constant weight was obtained. Subsequently, 3 g of Snow mountain garlic without shell were placed in the crucible. The sample was calcined in a Bunsen burner (model H-5875, Humboldt, CA, USA) under the hood until there was no release of vapors, then the sample was placed in a muffle for 2 h at 550 °C to put in a desiccator after that time. The crucible with the ashes obtained was weighed. The percentage of ashes was calculated with the formula: percentage of ashes = P − Q/W * 100, where *p* = weight of the crucible with the ashes in g; Q = weight of the empty crucible in g; W = sample weight in g. Each determination was made in triplicate.

#### 4.4.2. Acid Digestion of the Sample

100 mg of Snow mountain garlic ash was weighed using a Mettler Toledo brand analytical balance and was placed in a high-pressure Teflon cup. Later, Nitric acid grade ACS, J.T. Baker concentrate, was added to carry out the acid digestion process in a microwave oven (MARS 5 mark CEM). This analysis was done according to EPA 3015A [50].

#### 4.4.3. Quantitative Analysis by Inductively Coupled Plasma Emission Spectrometry (ICP)

It was made using a plasma spectrometer mark thermo model iCAP 6500 Series. The operating conditions of the ICP for the analysis were: radiofrequency power, 1150 W; assistant gas flow, 0.5 L/min; nebulizer gas flow, 0.60 L/min; wavelengths for each element (nm): potassium (766.4 nm); match (177.4 nm); magnesium (279.5 nm); calcium (393.3 nm); sodium (589.5 nm); zinc (213.8 nm); iron (259.9 nm); manganese (257.6 nm); copper (324.7 nm); arsenic (193.7 nm); cobalt (228.6 nm); nickel (231.6); plumb (220.3 nm); mercury (184.9 nm); selenium (196.0 nm); cadmium (226.5 nm). Ultra-pure grade argon was used. All ICP analyses were done according to EPA 3015A [50].

### 4.5. Extraction and Analysis of the Fatty Acid Composition

#### 4.5.1. Moisture Content and Volatile Compounds

The experiment was done in duplicate. It was carried out by the vacuum oven method. 5 g of garlic slices were placed in an oven at 65 °C for 4 h. The result was obtained in the percentage of evaporated material. Other volatile compounds also evaporated along with the water, but they were not identified for this work.

#### 4.5.2. Fat Content

It was determined by the Soxhlet method using hexane as solvent. The garlic sample (5 g) was used dryly in a filter paper cartridge, the Soxhlet system was connected, and the solvent was recirculated for 10 cycles. Finally, the solvent in the flask was evaporated to obtain the weight of the extracted fat sample. The experiment was done in duplicate [51].

#### 4.5.3. Fatty Acid Profile

The fatty acid profile of garlic oil was determined by gas chromatography by injecting the mixture of the methyl esters derived from the fatty acids of the samples. The methylation process consisted of reacting samples of 10 mg of oil with 0.5 mL of a 5% solution of sodium methoxide in methanol. The reaction was carried out by heating at 65 °C for 30 min and stopped by adding 0.5 mL of a saturated solution of NaCl. The methyl esters were extracted with 1 mL of hexane (HPLC grade) and subsequently analyzed on a Varian gas chromatography (model 3400) equipped with a split injection system set at 250 °C and a flame ionization detector set at 300 °C. A Stabilwax capillary column (Restek Corp.) of 60 m long × 0.25 mm and 0.25 µm stationary phase film was used. Hydrogen was used as a carrier gas. The column temperature was increased from 150 °C to 200 °C at a rate of 10 °C/min, then to 250 °C at a rate of 3 °C/min, and finally, it was kept at 250 °C for 20 min. The individual components were identified by comparison with the retention times of the fatty acid methyl esters of a standard mixture. Each sample was analyzed in duplicate. Each reported quantity corresponds to the percentage of each fatty acid concerning the 100% of fatty acids present in the sample [51].

### 4.6. Obtention of Extracts

#### 4.6.1. Preparation of the Sample to Obtain the Extracts

25 g of peeled Snow mountain garlic were weighed. Subsequently, it was cut into small pieces (approximately one mm) and placed in a drying oven (Felisa Fe-291ad) at 60 °C for 3 days to remove moisture. Once dry, it was crushed in a pestle and mortar until a fine powder was obtained [52].

#### 4.6.2. Continuous Extraction

The garlic powder was macerated in 250 mL of hexane (J.T. Baker^®^, Phillipsburg, NJ, USA) in an amber bottle for 3 days. After this time, it was filtered and transferred to a 1 L flask to evaporate the solvent on a rotary evaporator (BUCHI R-3). Subsequently, the extract was placed in a previously weighed vial labeled as hexanic extract. To the same garlic powder sample, 250 mL of chloroform was added (J.T. Baker^®^, Phillipsburg, NJ, USA). After three days, the sample was filtered and evaporated to obtain the extract (labeled as chloroformic extract). Finally, the same procedure was done with 250 mL of ethanol (J.T. Baker^®^, Phillipsburg, NJ, USA) to get the ethanolic extract [52].

### 4.7. Phytochemical Analysis

Each extract was dissolved in 2 mL of the solvent with which they were extracted. In a chromatographic plate and a different line, 3 drops of each extract were placed. Each chromatographic plate was developed in a glass chamber using different solvent systems to observe the separation of metabolites (chloroform/acetone 80:20 *v*/*v*, chloroform/ethyl acetate 90:10 *v*/*v*, ethyl acetate/hexane 80:20 *v*/*v*, ethyl acetate/methanol 80:20 *v*/*v*, chloroform/acetone 60:40 *v*/*v*, acetone/methanol 60:40 *v*/*v*, methanol/water 70:30 *v*/*v*, acetonitrile/water 70:30 *v*/*v*) [52].

#### 4.7.1. Determination of Saponins

##### Foam Test

2 mg of each extract was separately placed in a test tube, and 1 mL of distilled water was added. The tubes were capped and manually shaken for 4 min. This test recognizes the presence of both steroidal and triterpene saponins. The test is considered positive if foam appears on the liquid surface of more than 2 mm in height and persists for more than 2 min [53].

##### Libermann–Burchard Test

A sample of each extract was placed on chromatography plates. The plates were developed in the different systems mentioned and dried at room temperature to later submerge them in the Libermann–Burchard reagent (composed of acetic anhydride/sulfuric acid 19:1 *v*/*v* with 0.5 mL glacial acetic acid). The excess reagent was removed and left to dry at room temperature. A positive test presents a color change that starts from pink-blue (very fast), continues to bright green visible (fast), and ends in dark green-black, indicating the reaction’s end. This reaction is also used to differentiate the steroidal structures from the triterpene; the former produces colorations that range from blue to greenish-blue, while for the latter red, pink, or purple is observed [54].

#### 4.7.2. Determination of Alkaloids

A sample of each extract was placed on chromatography plates. The plates were developed in the different systems mentioned and dried at room temperature to subsequently submerge them in Dragendorff’s reagent (8 g de Bi(NO_3_) * 5 H_2_O in 20 mL de HNO_3_ (30%) + KI 27.2 g in 50 mL of water). The excess reagent was removed and left to dry at room temperature. A change in color, generation of precipitates, lumps, or flocculation indicate a positive result due to reaction with the amino group of the alkaloids [55].

#### 4.7.3. Determination of Sterols and Triterpenes

##### Salkowski’s Test

A sample of each extract was placed on chromatography plates. The plates were developed in the different systems mentioned and dried at room temperature to later place five drops of H_2_SO_4_ in each one and left to dry at room temperature. The test is positive if there is a yellow–red color change [56].

#### 4.7.4. Determination of Tannins

##### Ferric Chloride Test

A sample of each extract was placed on chromatography plates. The plates were developed in the different systems mentioned and dried at room temperature. They were immersed in 1% FeCl_3_, the excess reagent was removed, and they were left to dry at room temperature. The test is positive if it shows a color change depending on the type of tannins. The catechins give a green coloration while the gallic ones a blue coloration; if both are present, black coloration is obtained [52].

##### Vanillin Test

A sample of each extract was placed on chromatography plates. The plates were developed in the different systems mentioned and dried at room temperature. Five drops of HCl were placed and subsequently submerged in the 1% vanillin developer. The excess reagent was removed and left to dry at room temperature. The test is considered positive if a cherry red color appears [57].

##### Catechin Test

A sample of each extract was placed on chromatography plates. The plates were developed in the different systems mentioned and dried at room temperature to add five drops of HCl and put on a heating plate for 5 s, and finally, they were left to dry at room temperature. The test is positive if red coloration appears [56].

#### 4.7.5. Determination of Flavonoids

##### Constantinesco Test

A sample of each extract was placed on chromatography plates. The plates were developed in the different systems mentioned and dried at room temperature to submerge them in 10% sodium acetate and remove excess. Subsequently, they were immersed in AlCl_3_ solution, and the excess was removed and left to dry at room temperature. The test is positive if yellow coloration is observed [58].

##### Shinoda Test

A 0.5 mL sample of each extract was taken and placed in a separate test tube. Metallic Mg and 3 drops of HCl were added. The test is taken as positive if there is the presence of yellow, orange, brown, or red coloration [59].

### 4.8. Determination of Antioxidants

#### 4.8.1. Extraction Procedure of Polyphenols 

The extraction procedure was performed by the method used by Ovando-Martínez et al. (2009) [60] with some modifications. The sample (0.19 g garlic) was shaken for 1 h at room temperature with methanol: water acidified with HCl solution (70:30 *v*/*v*, pH 2, 20 mL/g sample) and centrifuged (15 min, 10 °C, 11,000 rpm), supernatants were collected, and precipitates were resuspended in an acetone: water solution (70:30 *v*/*v*, 20 mL/g samples, 60 min, room temperature). After centrifugation (15 min, 10 °C, 11,000 rpm), supernatants were combined to determine the total phenolic content and antioxidant capacity.

#### 4.8.2. Determination of Total Phenolic Compounds

The total phenolic content (TPC) of compounds in the garlic extracts was determined via the Folin–Ciocalteu method [61], using gallic acid as the standard. The reaction was carried out with 100 µL of the extract, 100 µL of Folin–Ciocalteu reagent diluted (1 N), 2.0 mL of distilled water, and 2.8 mL of a solution of Na_2_CO_3_ (7.5%). The absorbance was measured at 750 nm. The TPC was expressed as the gallic acid equivalent (GAE) per mass of dry garlic (mg GAE/g dw). The full equivalence values were calculated using the standard curve of gallic acid.

#### 4.8.3. Determination of Antioxidant Activity by DPPH Radical

The extract samples were measured in terms of hydrogen-donating or radical-scavenging ability using the stable DPPH radical [62]. The reaction mixture contained 100 µL of the extract and 3.9 mL of DPPH, and the absorbance of the reaction mixture was measured at 515 nm against a blank sample containing only methanol. The results were expressed in terms of the mass of Trolox per mass of dry garlic (µmol of Trolox Equivalent/g dw), and the full equivalence values were calculated using the standard curve of Trolox.

### 4.9. Statistical Analysis

The experiments were done in duplicate or triplicate, indicating the standard deviation (SD).

## 5. Conclusions

Snow mountain garlic shares a composition similar to those found in other garlic. The results indicate that the presence of phenolic compounds in Snow mountain could be responsible for the pharmacological properties. Besides, the phytochemical analysis showed secondary metabolites such as saponins, alkaloids, tannins, and flavonoids; these metabolites also contribute to Snow mountain benefits. Further studies are necessary to understand better the role of fatty acids in the metabolism and structure of garlic, the effect of antioxidants, and the conservation of properties when storing or cooking it.

## Figures and Tables

**Figure 1 molecules-27-03712-f001:**
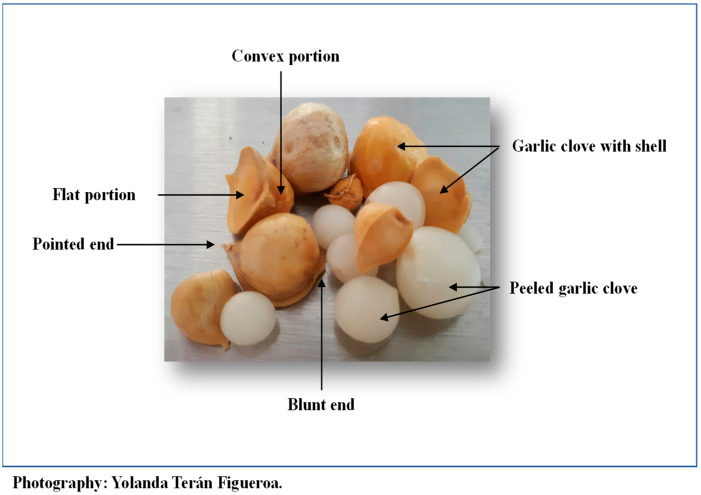
Description of Snow mountain garlic.

**Table 1 molecules-27-03712-t001:** Determination of units and measures of Snow mountain garlic cloves found in 100 g.

Units	Size	Height (cm)	Width (cm)	Base (cm)
41	Big	1.3–1.5	1.5–2.0	1.5–2.5
65	Medium	0.7–1.2	1.0–1.2	1.2–1.5
37	Small	0.8–0.9	0.7–0.8	0.7–1.0

Note: 100 g were weighed separately in 3 groups. The units and measures are the results of the average of the 3 groups.

**Table 2 molecules-27-03712-t002:** Snow mountain garlic bromatological analysis and crude energy and pH.

COMPONENT	
Humidity (%)	61.5628 ± 0.5411 ^a^
a_w_	0.9817 ± 0.0081 ^a^
Carbohydrates (%)	46.7 ± 0.0024 ^b^
Lipids (%)	0.5 ± 0.0036 ^b^
Protein (%)	0.5 ± 0.0029 ^b^
Ashes (%)	1.2 ± 0.004 ^b^
Crude energy (kcal/100 g)	193.3 (410.7 in *A. sativum* L.)
pH	6.5 ± 0.002 ^b^

^a^ average of 3 independent experiments. ^b^ average of 2 independent experiments.

**Table 3 molecules-27-03712-t003:** Snow mountain garlic aches chemical analysis.

Element	Concentration (g/kg)	Element	Concentration (mg/kg)
Potassium (K)	287.46	Copper (Cu)	84.80
Phosphorus (P)	91.58	Arsenic (As)	<LDM
Magnesium (Mg)	21.20	Cobalt (Co)	<LDM
Calcium (Ca)	6.16	Nickel (Ni)	<LDM
Sodium (Na)	1.04	Plumb (Pb)	<LDM
Zinc (Zc)	0.51	Mercury (Hg)	<LDM
Iron (Fe)	0.42	Selenium (Se)	<LDM
Manganese (Mn)	0.31	Cadmium (Cd)	<LDM

<LDM = less than the limit of detection of the method.

**Table 4 molecules-27-03712-t004:** Fatty acid profile of Snow mountain garlic essential oil.

Fatty Acid	Name	% (SD)
C10:0	Decanoic acid	5.00 (0.41)
C13:0	Tridecanoic acid	5.34 (0.32)
C15:0	Pentadecanoic acid	3.54 (0.09)
C16:0	Hexadecanoic acid	4.22 (0.65)
C16:2 or C16:3	Hexadecadienoic acid or Hexadecatrienoic acid	53.50 (3.56)
C18:0	Octadecanoic acid	2.05 (0.25)
C18:1	Octadecenoic acid	3.10 (0.22)
C18:2	Octadecadienoic acid	10.93 (1.10)
C18:3	Octadecatrienoic acid	1.69 (0.20)
C20:1	Eicosanoic acid	2.19 (0.21)
C20:2	Eicosadienoic acid	3.44 (0.18)
C20:3	Eicosatrienoic acid	5.00 (0.47)
Saturated		20.15
Monounsaturated		4.24
Polyunsaturated		75.61

SD = standard deviation.

**Table 5 molecules-27-03712-t005:** Result of phytochemical analysis on Snow mountain organic extracts.

Metabolite Type	Reaction	Extracts		
		HxE	CHCl_3_E	EtE
Saponins	Foam	+	++	+++
	Libermann–Burchard	+	++	+++
Alkaloids	Dragendorff	−	++	+++
Triterpenes	Salkowski	−	+	+
Tannins	Ferric chloride	+	+	+
	Vanillin	+	+++	+++
	Catechin	−	+	++
Flavonoids	Constantinesco	−	++	+++
	Shinoda	−	+	+++

HxE, hexanic extract; CHCl_3_E, chloroformic extract; EtE, ethanolic extract. “+”: (positive), “−”: (negative).

## Data Availability

Not applicable.

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
