# Peer review of "Bromatological Analysis and Characterization of Phenolics in Snow Mountain Garlic"

_molecules, 2022, doi:10.3390/molecules27123712_

Round 1

Reviewer 1 Report

English has to be improved, there are some grammar and spelling mistakes.

I suggest to include a Latin name of the Snow mountain garlic in a title of the manuscript.

The abstract could be written more precisely, f.ex. “a higher percentage” or “higher amount of polyunsaturated (75.61%)” we do not know to what we should compare it. Higher than what? Also when you write about the percentage it is not an amount.

A keyword “Snow mountain” is not appropriate to this study.

As the fat content is only 0.5% I do not find it very useful to analyze fatty acid profile. It may differ significantly depending on the sample. Additionally it has a little influence on the nutritional value.

Lines 60-62: provide some numbers and specify the compounds based on the literature.

Line 67: “Snow mountain garlic produces a single clove”. Is “produces” a right word in this context?

In the bromatological analysis the water content would be more precise, rather than, “humidity”.

Table 1 such a measurement of the sample of 100 g of the material is not representative.

Statistical analysis is missing. Provide at least standard deviations in the tables 2 and 3. In the Methods information about the statistical analyses should be included.

Material: “The Snow mountain garlic was obtained from the local market” this information is not enough. Please provide the region and year of a cultivation at least.

Provide information about the methods of humidity and ashes content. Which temperatures did the Authors apply? How the ashes were calculated?

Point 4.5.1. the point about the volatile compounds is not clear. What exactly and how was measured?

A lot of compounds and parameters were measured but the discussion and conclusion is very short. In my opinion the manuscript should focuses only on selected and most important parameters and compounds.

The conclusion is very poor. I suggest to delete everything what was written in the Conclusion and write it once again. In this version of the manuscript I do not see any important conclusions.

Author Response

Annex corrections 

Thank you 

Reviewer 2 Report

This manuscript describes some analyses of snow mountain garlic, such as: bromatological analysis, lipid profile, ash content and a general characterization of some phenolics’ groups.

In general, the analyzes are very general and unspecific, thus not being able to differentiate this garlic from others. And, in my opinion, this is a problem, because, according to the authors, this is the differential and new of the work.

The main analyzes of the bioactive compounds of garlic are the characterization of specific phenolic compounds, such as: profile of flavonoids, alkaloids, steroids, triterpenes, sulfur compounds, etc. This study only brings general analyzes that only demonstrate the presence of some groups of bioactive, but this is not innovation, and these are very superficial results for differentiation and characterization. Thus, I guess this manuscript could only be accepted if the authors brought these results, so that they could really differentiate this garlic from others and characterize it.

Then, according to me, however this manuscript brings some interesting results, these results are not sufficient to publish in Molecules. Maybe this results can be published as a short communication. 

However, if the Editor understands that this manuscript can be published in Molecules, I have some recommendations:

  • Title:

The title does not reflect the main analyzes were made, nor is it in accordance with the proposed objective.

  • Introduction:

The introduction can be improved. According to me is some superficial.

  • Material and Methods:

4.1: This information is so important, then, the authors should be more specific: where? when? Was the sample identified by an expert?

4.4.2; 4.5.3; 4.6.1; 4.6.2; 4.7.; 4.7.1.3: In these items the reference should be added. If the authors development the method, the data of this validation/development should be added.

The research design and the statistical analysis used by authors should be added.

  • Results:

In general, this section is good and describes the results of the study.

Table 4: I think is interesting added the name of the fatty acids.

  • Discussion:

This part can be improved. The discussion about the minerals and the lipid profile can be improved (the importance of this results, and of this compounds).  

  • Conclusion: In my opinion should be improved, because doesn’t reflect the main results and doesn’t response of the aim of this study.

Best regards.

Author Response

Annex corrections 

Thank you 

Reviewer 3 Report

The Authors carried out a qualitative analysis of the metabolites present in the extracts studied and a quantitative analysis in relation to the protein, fat or mineral content of the raw material studied. Generally, the studies were well planned by the Authors. Several points need clarification.

Remarks:

  1. The Authors, pointing to the broad spectrum of biological properties of garlic, should provide two other references. (Lines 44-46)
  2. What does the description given by the Authors in subsection 4.5.2 of the manuscript refer to? Do the Authors give the fat content of the material tested or the yield of the extraction performed?
  3. In the procedure describing the extraction of polyphenols, the following wording needs clarification: What constituted "the sample" subjected to extraction? Were the aqueous-methanol and aqueous-acetone supernatants combined to determine the total polyphenol content?
  4. In which units was the antioxidant activity of the tested extracts determined? Was it really "the mass of Trolox per mass of dry powder juice"?
  5. Journal names should be standardised as required. (References)
  6. The entire text of the manuscript should be carefully reviewed, as there are sometimes missing letters or letter substitutions in a word for example “gran-negative" or “Tripertenes”.

Author Response

Annex corrections 

Thank you 

Round 2

Reviewer 1 Report

Thank you for improving your article and the replies to all my comments and suggestions. In my opinion, the paper in its current form is suitable for publication.

Reviewer 2 Report

Dear Editor,

The manuscript has been improved substancially.  Then, can be published. 

Best regards.